# Gated Path Aggregation Feature Pyramid Network for Object Detection in Remote Sensing Images

**Yuchao Zheng** [1,2]**, Xinxin Zhang** [1,2,*]**, Rui Zhang** [1,2] **and Dahan Wang** [1,2]

1  College of Computer and Information Engineering, Xiamen University of Technology, Xiamen 361024, China
2  Fujian Key Laboratory of Pattern Recognition and Image Understanding, Xiamen University of Technology, Xiamen 361024, China
*  Correspondence: zhangxinxin@xmut.edu.cn

**Abstract:** Object detection in remote sensing images is a challenge because remote sensing targets have characteristics such as small geometries, an unfixed direction and multiple poses. Recent studies have shown that the accuracy of object detection can be improved using feature fusion. However, direct fusion methods regard each layer as being of equal importance and rarely consider the hierarchical structure of multiple convolutional layers, leading to redundancy and rejected information being rarely applied during the fusion process. To address these issues, we propose a gated path aggregate (GPA) network that integrates path enhancement and information filtering into an end-to-end integrated network. Specifically, we first quantitatively analyze the performance of different gating functions to select the most suitable gating function. Then, we explore the embedding of soft switchable atrous convolution (SSAC) in the topmost feature layer. Finally, we validate our proposed model by combining it with experiments using the public NWPU VHR-10 dataset. The experimental results show that our proposed GPAFPN structure has significant improvement compared to the FPN structure. Compared with the mainstream networks, it achieved state-of-the-art performance.

**Keywords:** feature aggregation; object detection; remote sensing images

## 1. Introduction

Object detection in high-spatial-resolution (HSR) remote sensing images has been widely applied in many areas, such as environmental supervision, military, transportation, urban monitoring and management. With the increasing spatial resolution of remote sensing images and the great variation in object size in said images, the optimal selection of the spatial region window size is usually difficult to determine and needs to be adjusted according to the actual scenario. These features lead to an imbalance in the distribution of foreground and background context and diverse small objects along with the complex background [1]. Effective extraction of semantic information can alleviate these problems. Therefore, effectively extracting contextual information is a hot research topic.

Currently, mainstream research areas include small target detection [2], dense detection [3], rotating target detection [4–8] and multiscale feature extraction combination [9–13]. These methods expect to discover a general network architecture with fewer parameters and better expressiveness. Many studies have indicated that lower-feature layers have better localization information, while higher-feature layers have rich semantic information. Thus, the combination of multiscale extraction has high research value since it can integrate different feature maps to achieve complementary advantages. This paper focuses on how to better combine multi-scale feature extraction, and proposes solutions to the two problems of redundant noise information generated during feature fusion and target missed detection.

Fusion methods include addition, multiplication and splicing. There are many methods of connection fusion regarding feature maps, such as direct connection, selective connection and circular connection. A well-known fusion method is adaptive spatial feature

fusion (ASFF) [14], which utilizes an attention mechanism to learn different feature maps and obtain different feature contributions. A PAFPN [15] shortens the information channel from the bottom to the top by adding an additional bottom-up path. A recursive-FPN [16] not only cycles the FPN [17] structure twice but also embeds atrous spatial pyramid pooling (ASPP) [18] in the connection, while part of the convolution adopts switchable atrous convolution (SAC) so that the image-level information is embedded into the feature map.

Multiscale feature analysis is a useful strategy for better learning target features. Shallow larger-scale features are suitable for dealing with small targets because they have a smaller sensing field, higher resolution and stronger geometric detail information characterization ability, but their semantic information characterization ability is insufficient. On the other hand, deeper small-scale features have a larger perceptual field and stronger semantic information characterization ability, but their feature map has lower resolution and weaker geometric information characterization ability (lack of spatial geometric feature details). Common multiscale feature processing is divided into structure and fusion methods.

The FPN is one of the most classic and widely used multiscale feature fusion structures. It consists of a lateral connection and a top-down pathway, merging by addition. Based on an FPN, PANet uses the addition of a lateral connection and a down-top pathway to reduce the distance between the bottom and top layers. BiFPN [19] improves PAFPN by adding an original layer P4-P6 and a lateral layer to form a BiFPN layer and then serially repeats the computation three times. NAS-FPN [20] applies reinforcement learning, which is equivalent to a merit-seeking algorithm, to select the best model structure in a given search space. The reward signal is the accuracy of the submodules in the search space. A recursive-FPN loops the FPN structure twice, and a variant atrous spatial pyramid pooling (ASPP) is added between the FPN loops.

A fusion method uses operations such as addition, multiplication and splicing when merging feature layers of different scales. ASFF, applied in the famous YOLOv3, simultaneously uses addition and multiplication. Similar to a full connection, the brief idea of ASFF is to add a learnable coefficient on the basis of the original FPN. The learning coefficient is automatically learned; hence, an adaptive fusion effect can be achieved. Various modules, such as attention mechanisms, can be embedded when connecting different feature layers. Squeeze-and-excitation (SE) [21] is widely used, in which the first step is compression, achieved through global average pooling, the second step is excitation, in which two fully connected layers are followed by an activation function, and the last step is a scale operation. However, SE dimensionality reduction causes side effects to channel attention prediction, and the dependence between channels is inefficient and unnecessary. To address the two shortcomings of SE, the ECA [22] module is proposed. This module does not have a dimensional reduction process. Instead, it directly captures local cross-channel interactions by following k-nearest neighbors directly after global average pooling. Therefore, it can effectively capture cross-channel interactions. A convolutional block attention module (CBAM) [23] includes a channel attention module (CAM) and spatial attention module (SAM). The CAM has one more parallel global average pooling branch than SE. The SAM performs concatenation operations on the results of global max pooling and global average pooling and then obtains the spatial attention feature through dimensionality reduction and activation functions.

Nevertheless, these multiscale feature fusion methods all face the problem that the fused features have considerable redundant and mutually exclusive information, and the different feature layers span a large distance from the bottom layer to the top layer [24]. If there is no bottom-up path, the shallow information (P2) to the deep information (P5) has approximately tens to hundreds of layers in ResNet, which will cause serious loss of shallow information and reduce detection accuracy. Some objects that need to be detected in remote sensing images, such as storage tanks, ships and vehicles, occupy less than 100 pixels on average, which is too small for a $1300 \times 800$ image. For example, in the ResNet, these targets are downsampled by 16 times to correspond to the P5 layer and is about 0.4 pixels. The traditional P6 is generated by performing a max-pooling operation on

P5, which is easily replaced by other features with the max-pooling operation. Simply using a convolution with a stride of two can also downsample by a factor of two, but it cannot guarantee that the receptive field remains unchanged. Atrous convolution can ensure that the receptive field remains unchanged without increasing the cost of calculation. In this study, a proposed GPANet adopts a dual-path structure with a channel attention mechanism to strengthen useful features and replace maxpooling with SSAC avoids sudden reduction of receptive field and enhances object detection ability.

The main contributions of this paper are summarized as follows.

- Aiming at the problem of redundant noise information generated by PAFPN in the feature fusion process, we use an additional gating function, namely GPAFPN. The experiments are conducted to explore which gating function is more suitable.
- For the problem of missed target detection, replacing maxpooling with SSAC structure can not only prevent sample features from being replaced by other features, but also ensure that the receptive field and computational cost remain unchanged. The experiments explore which feature layer is the most appropriate to derive the top layer.
- A quantitative analysis with the mainstream attention mechanism on the NWPU VHR-10 [25] dataset is performed and the optimal mechanism is selected. Comparing with a series of popular networks, our proposed GPANet shows significant improvement.

## 2. Materials and Methods

For various detection targets, the features from different convolutional layers are different. The semantic features in the top layer are more suitable for target classification, while the detailed features in the bottom layer are more suitable for target localization. Figure 1 shows how the FPN and PAFPN structures use multiscale information fusion to achieve the fusion of low-level information and high-level information; hence, more contextual information is available at each layer. This section focuses on two important modules of GPAFPN and SSAC. The next section describes the data enhancement approach and ablation experiments.

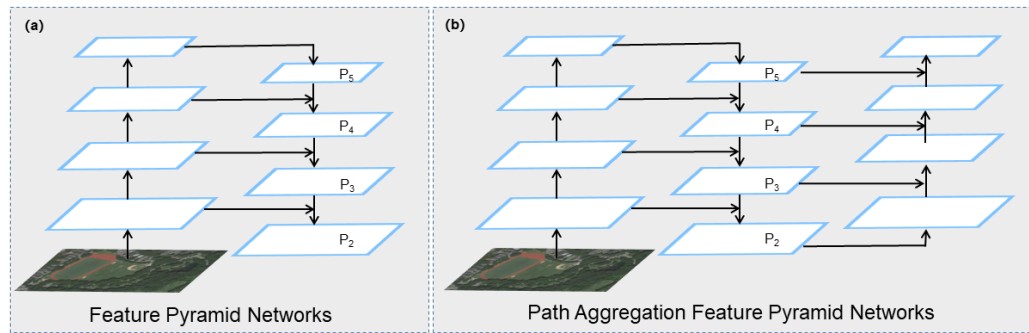

**Figure 1.** (**a**) Structure diagram of FPN and (**b**) Structure diagram of PAFPN. Comparing the two, PAFPN has one more bottom-up path than FPN.

With the advantage of FPN-based structure, we propose a GPANet framework for object detection of remote sensing images. The framework is shown in Figure 2, the proposed GPANet is based on the Faster RCNN model, which consists of five modules: (a) FPN feature extractor, (b) bottom-up gated connection, (c) regional proposal network (RPN), (d) ROIPooling and (e) Faster RCNN head [26]. (a) ResNet-34 or ResNet-50 [27] is used to extract the conv2_x, conv3_x, conv4_x and conv5_x layers. (b) is based on (a) with an attention mechanism and a feature map fine-tuned by SSAC. (c) The region of interest is screened and the context is evaluated. (d) Features of the same size are obtained through ROIPooling. (e) Contains two branches: classification uses two fully connected layers and softmax to calculate category confidence and positioning uses bounding box regression to obtain the position offset of each proposal, which is used to return to a more accurate target detection frame. As seen from the overall pipeline, our GPANet is an improvement of the

Faster RCNN model. In brief, the main difference between GPANet and Faster RCNN is that GPANet combines a GPAFPN structure, i.e, module (b), with the Faster RCNN model. Compared with the FPN structure, the advantage of GPAFPN is it can make the features more representative and improve the utilization of the P6 layer by adding a bottom-up gating function and an SSAC structure to the original FPN.

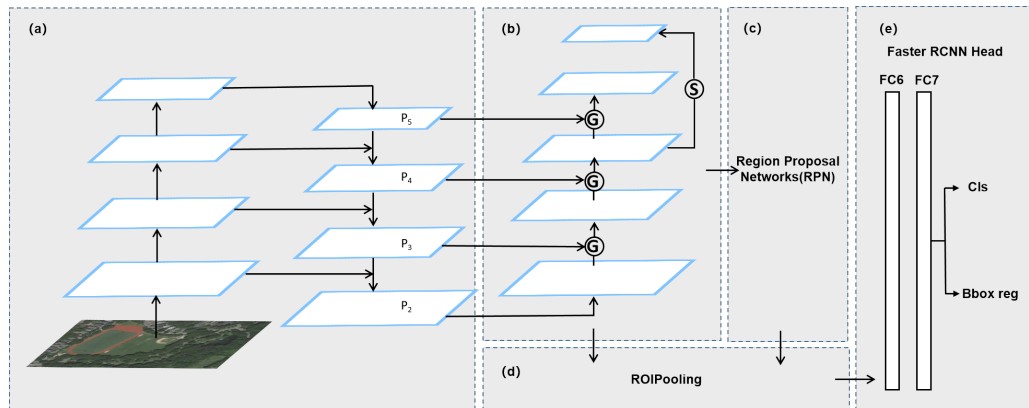

**Figure 2.** The overall pipeline of the proposed GPANet. (**a**) FPN feature extractor, (**b**) bottom-up gated connection, (**c**) regional proposal network (RPN), (**d**) ROIPooling and (**e**) Faster RCNN head.

## 2.1. Soft Switchable Atrous Convolution Module

In this section, SSAC will be introduced, which is modified from the original SAC. Figure 3 shows the overall architecture of SSAC. It is composed of three parts: two global context components before and after, and a SSAC component is inserted in the middle. These two components will be analyzed in detail next.

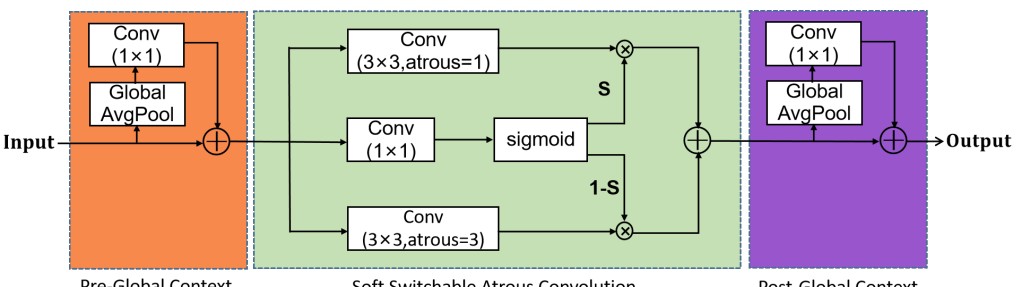

**Figure 3.** SSAC framework.

### 2.1.1. Atrous Convolution

Atrous convolution is the operation of inserting atrous into a standard convolutional map to increase the perceptual field. Compared with the standard convolution, it has an additional hyperparameter dilation rate, which refers to the number of intervals of the kernel with a weight of 0. If the standard convolution kernel size is $k \times k$ and the atrous rate is r, then the atrous convolution kernel is $k_d = k + (k-1)(r-1)$. For example, the standard convolution kernel size shown in Figure 4 is $3 \times 3$, indicating that the atrous convolution has atrous rates of 1 (top) and 2 (bottom), which means that the void convolution can map similar targets of different sizes with different void rates, and the computational effort remains constant.

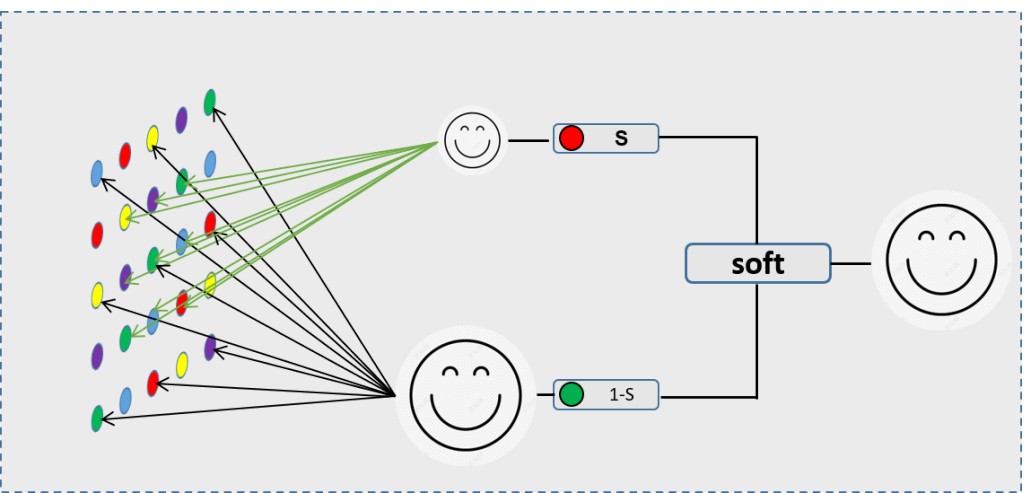

**Figure 4.** SSAC intuitive map, including convolution with different atrous rates.

### 2.1.2. Soft Switchable Atrous Convolution

The convolution manipulation can be expressed as $y = Conv(x, w, r, k)$ with atrous rate $r$ and kernel size $k$, which takes $x$ as its input and outputs $y$. The specific formula for converting the standard convolution manipulation to SSAC is shown as follows.

$$Conv(x, 1, k) \xrightarrow{Convert\ to\ SSAC} S(x) \cdot Conv(x, 1, k) + (1 - S(x)) \cdot Conv(x, r, k) \quad (1)$$

$$S(x) = Sigmoid(Conv(x, 1, k = 1)) \quad (2)$$

In Equation (1), $r$ is a hyperparameter, $r$ and $k$ are both 3 in this study. The S-function is composed of a $1 \times 1$ convolutional layer and a sigmoid function, and its input is the result of the pre-global context (see Figure 3). The soft mechanism can determine the weights of two atrous convolutions, which is somewhat similar to adaptive convolution. Compared with the original SAC, a SSAC cancels the locking mechanism and changes the rigid selection of the S-function into a flexible one. The advantage is that the void rate can no longer only be selected from two, that is, the receptive field is no longer fixed, which is more conducive to target detection.

### 2.1.3. Global Context

The GPAFPN structure is to add a gating function on the bottom-up path on the basis of PAFPN. A global context module is located at both ends of the SSAC module, and it is a lightweight module. The module is mainly inspired by SE. The difference between the global context module and the squeeze-and-excitation module is that the former has only one convolutional layer, while the latter is a perceptron. The former adds the processed output back to the mainstream by addition, while the latter uses multiplication. The global context module consists of global average pooling and a $1 \times 1$ convolution. The computed result is then multiplied by the original input to ensure that the sizes of the output feature and input feature are equal. The global context module for S-function acts as a stable prediction.

### 2.2. Bottom-Up Gating Function

A bottom-up gating function is added between adjacent layers on the bottom-up path to make full use of the complementary information of the adjacent layer, and it also reduces noisy information during the fusion process. The bottom-up gating function adopts an attention mechanism. The initial layers are P2-P5, obtained using the ResNet network as the backbone and an FPN as the extractor. The feature layers are all of different sizes, so a downsampling operation is adopted during the initial fusion, which is implemented

by a $3 \times 3$ convolution with a step size of 2. Figure 5 shows the detailed structure of the bottom-up gating function.

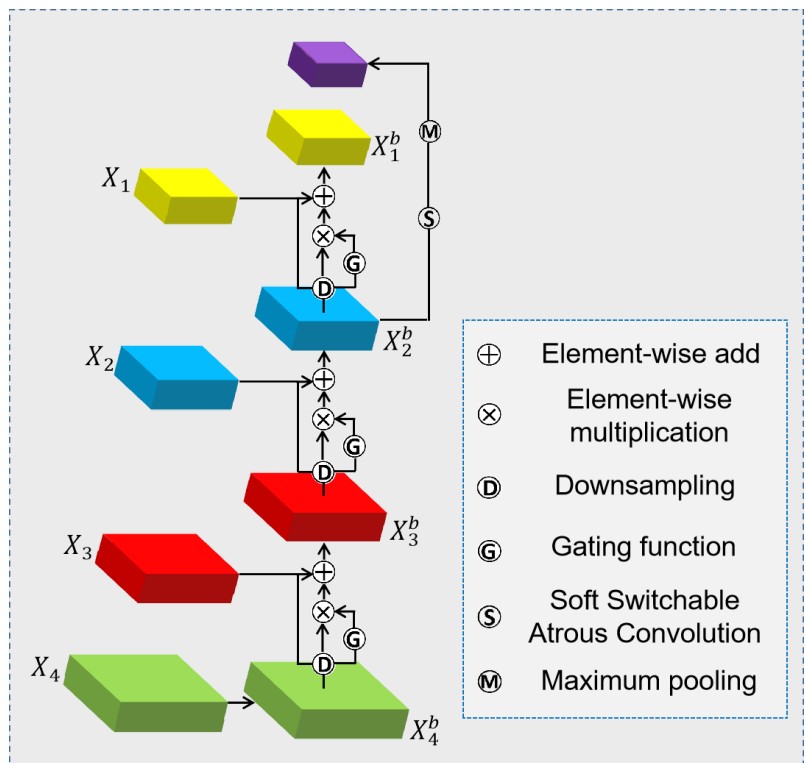

**Figure 5.** Framework of the bottom-up gating function.

Standardized ResNet extracts the conv2_x, conv3_x, conv4_x and conv5_x layers and then uses the FPN to obtain $X_1 \in R^{M_1 \times N_1 \times C}$, $X_2 \in R^{M_2 \times N_2 \times C}$, $X_3 \in R^{M_3 \times N_3 \times C}$ and $X_4 \in R^{M_4 \times N_4 \times C}$, respectively. The number of channels in each layer is the same, and the length and width between $X_1$ and $X_4$ are all scaled by 0.5 times. The feature layer at the bottom has detailed positioning information. By merging from the bottom to the adjacent upper layer, the semantic information of the upper layer can be strengthened. Note that $X_4$ is simply $X_4^b$, without any processing.

First, $X_4 \in R^{M_4 \times N_4 \times C}$ performs a downsampling operation, which is specifically achieved by $3 \times 3$ convolution with a step size of 2, obtaining $X_4^b\_D \in R^{M_3 \times N_3 \times C}$. The gating function controls the transfer of information from $X_4^b\_D$ to $X_4^b\_G$. The gating function is inspired by the SE module and generates a C-dimensional gating vector $g_4^b\_D \in R^{1 \times 1 \times C}$ (where C is the channel number of $X_4^b\_D$) with values in the range 0 to 1. Multiplying the *i*-th channel of A with the *i*-th element of the corresponding B serves to strengthen the effective channel and suppress the ineffective one. The gating function consists of global average pooling and the simplest multilayer perceptron (MLP) with a ReLU function. $X_4^b\_D$ is first generated as a $1 \times 1 \times C$ vector by a global average pooling layer, and then $g_4^b$ is generated by the simplest multilayer perceptron. $g_4^b$ can be formulated as:

$$g_4^b = Sig(fc(\sigma(fc(pool(X_4^b\_D)))))$$ (3)

Here, $Sig(\cdot)$ denotes the Sigmoid activation function, $Sig(x) = \frac{1}{1+\exp^{-x}}$, $fc(\cdot)$ denotes the fully connected layer, $\sigma(\cdot)$ denotes the ReLU function, and $pool(\cdot)$ denotes the global average pooling layer. With the gating function, the information can be added to the next layer after noise reduction. The information source of $X_3^b$ consists of three parts, including $X_3$ after convolution, $X_4^b\_G$ after gating function and $X_4^b\_D$ after downsampling. $X_3^b$ can be formulated as:

$$X_3^b = X_3 + X_4^b\_G + X_4^b\_D$$ (4)

$$X_4^b\_G = g_4^b \cdot X_4^b\_D \tag{5}$$

Here, $\cdot$ represents the $i$-th element of $g_4^b$ multiplied by the $i$-th channel of $X_4^b\_D$. Part $X_3$ is used as the original bottom-level information. Part $X_4^b\_G$ is used for screening in the process of information transmission. Part $X_4^b\_D$ is equivalent to adding a jump connection. The main purpose is to turn the original information filtering operation into an information enhancement operation, because the original underlying information is used as the basis. Similarly, the calculation formulas of $X_2^b$ and $X_1^b$ are as follows:

$$X_j^b = X_{j+1}\_G + X_{j+1}^b\_D \quad , j = 1, 2 \tag{6}$$

$$X_{j+1}^b\_G = g_{j+1}^b \cdot X_{j+1}^b\_D \quad , j = 1, 2 \tag{7}$$

$$g_{j+1}^b = Sig(fc(\sigma(fc(pool(X_{j+1}^b\_D))))) \quad , j = 1, 2 \tag{8}$$

Here, $X_{j+1}^b\_D$ is obtained by down-sampling from $X_{j+1}^b$ respectively, ensuring that the feature shapes are all the same size.

### 2.3. Top-Level Features after Fine-Tuning

In the classic PAFPN structure, the top-most feature is obtained using a maximum pooling operation with a step size of 2. In this experiment, the method of obtaining the top-level features has been changed, and maximum pooling operation is replaced by SSAC operation. The purpose of adding the SSAC operation is to adaptively fuse two atrous convolution operation. So that a certain target can still be detected in the topmost special layer. Targets of different sizes are adapted to feature maps of different scales, and fine-tuning the top-level operation is equivalent to direct expansion of this feature maps.

## 3. Experiments and Analysis

This section first presents the dataset for the experiments and the details of the proposed GPANet. The effect of adding SSAC and bottom-up gating functions on the model is analyzed through ablation experiments. Next, a series of gating functions are quantitatively analyzed to select the best pairing for the model. Finally, experiments are conducted at different ResNet depths to observe whether the proposed module is robust. At the same time, compared with the current models that have been run on this dataset, the method proposed in this paper has obvious improvement effect.

### 3.1. Dataset Description

Compared with natural image datasets, such as ImageNet and Microsoft Common Objects in Context (MS COCO), which have hundreds of thousands—or even millions—of labeled images, the amount of available remote sensing datasets for object detection is only a few hundred to a few thousand. Therefore, the image enhancement techniques are needed to expand the sample size and reduce the effects of sample imbalance. The experiment used the NWPU VHR-10 [25] dataset, which was annotated by Northwestern Polytechnical University. The dataset is divided into two categories. There are 650 images that contain annotation information and 150 images that do not, for a total of 800 images. The experiment uses the 650 images with annotated information, and the detailed information of the content is shown in Table 1. There are a total of ten categories: airplane, ship, storage tank, baseball diamond, tennis court, basketball court, ground track field, harbor, bridge and vehicle. Among them, 715 pictures are from high-resolution remote sensing images captured by Google Maps, with a spatial resolution of 0.5 to 2 m, and the remaining 85 are panchromatic sharpened-color infrared images from Vaihingen data, with a spatial resolution of 0.08 m.

**Table 1.** Target number table of datasets before and after enhancement.

| Category | Object Size (pixel) | Number of Original Objects | Enhancement Method | Number after Enhancement |
|---|---|---|---|---|
| airplane | 33–129 | 757 | RFC | 741 |
| ship | 40–128 | 302 | RFC | 741 |
| storage tank | 34–103 | 655 | GC | 663 |
| baseball diamond | 49–179 | 390 | FG | 862 |
| tennis court | 45–127 | 524 | RF | 636 |
| basketball court | 52–179 | 159 | FG | 846 |
| ground track field | 192–418 | 163 | FG | 732 |
| harbor | 68–222 | 224 | RF | 640 |
| bridge | 98–363 | 124 | FG | 696 |
| vehicle | 42–91 | 598 | RFC | 591 |
| Total of the target quantity | – | 3896 | – | 7148 |

Due to the imbalance in the number of samples in each category, the predicted results themselves are category-oriented, so a specific data enhancement method is used to achieve the balance of the number of samples in each category. The image enhancement method in this study selects several of the four methods for combination, and Table 2 describes the methods in detail. *R* (rotation) represents the origin of the image center, anticlockwise rotating 90 degrees, 180 degrees and 270 degrees, *F* (flip) represents vertical or horizontal mirror flip, *G* (Gaussian blur) represents Gaussian filtering, $\sigma = (0, 0.5)$ and *C* (cropping) represents the removal of 0 to 10 pixels on any side of the periphery. To balance the data categories, the number of repeated operations performed on each picture is different (including a picture containing multiple categories), and the final number of different categories is approximately the same. The specific information of each category and the information after the enhancement change are shown in Table 1. After the image enhancement process, the total number of objects has been expanded to 7148, and we randomly selected 80% of the enhanced dataset as the training and the remaining 20% as the test dataset. The subsequent model accuracy comparison is based on the test samples.

**Table 2.** Dataset enhancement detail table.

| Enhancement method | Parameters |
|---|---|
| R (Rotation) | $Angle = [90, 180, 270]$ |
| F (Flip) | Horizontal or vertical |
| G (GaussianBlur) | $Sigma = (0, 0.5)$ |
| C (Cropping) | $Pixel = (0, 10)$ |

*3.2. Experimental Training Details*

Table 3 shows the experimental training parameters and anchor generation rules. We use the P2-P5 extracted by ResNet-50 or ResNet-34 as the input to the GPAFPN, and then the results go through the RPN network and finally utilize the Faster RCNN head as the detection head for classification and prediction. The neck takes advantage of the FPN as the baseline. The anchor sizes in the anchor generation strategy are 4 and 8. The aspect ratio is [0.5, 1, 2], and the step size is [4, 8, 16, 32, 64] (corresponding to the downsampling multiple; otherwise, it will result in no anchor in part of the images or the anchor setting will be beyond the edge of the images). To avoid the size of the input image being too large or excessive compression, the experiment set the border of the input image from 800 to 1333 pixels, which is close to the pixel size of the original dataset. The initial learning rate is 0.0025, and there are 12 epochs in total. The Linear warmup strategy is used for the 8th and 11th times. The learning rate increases linearly from a very small value to a preset value and then decreases linearly.

**Table 3.** Experimental training parameters and anchor generation rule table.

| Parameters | Value |
| --- | --- |
| Learning Rate | 0.0025 |
| Image Resize | $[800, 1333]$ |
| Momentum | 0.9 |
| Weight Decay | 0.0005 |
| Anchor Size | $[4, 8]$ |
| Ratios | $[0.5, 1, 2]$ |
| Strides | $[4, 8, 16, 32, 64]$ |
| Warmup | Linear |
| epochs | 12 |
| Batch size | 2 |

*3.3. Evaluation Criteria*

The experiment uses the COCO evaluation criteria and the average precision (AP) to evaluate the final experimental performance. AP: To explain this criterion, some prior knowledge is needed. First, TP, FP and FN are used to represent the numbers of true positives, false positives and false negatives, respectively, and the formulas for calculating precision and recall are as follows:

$$Precision = \frac{TP}{TP + FP} \tag{9}$$

$$Recall = \frac{TP}{TP + FN} \tag{10}$$

When the IoU value of the test result and ground truth is greater than the threshold, the test result is a true positive; otherwise, it is a false positive. According to precision and recall, the PRC curve can be drawn, where AP is the integral of the curve. The mAP is the average value of AP in each category. In the COCO data evaluation standard, the AP is the mAP. There are different AP values for different IoU benchmarks. There are three types of default IoU. The first is 10 IoU thresholds of 0.50:0.05:0.95. It is the most stringent evaluation index among the three, and is equivalent to the COCO evaluation index. The second indicator, the IOU threshold, is 0.5, which is the most relaxed indicator, and this is the evaluation indicator of other models in this paper. The third metric, the IOU threshold, is 0.75, and this metric is between the above-mentioned levels of stringency. The AP used in this experiment is the first standard.

**4. Discussions**

All experiments in this paper are implemented through the MMDetection [28] framework. To evaluate the significance of the added module to the experiment, ablation experiments are performed on the NWPU VHR-10 dataset. The ablation experiment does not modify other contents of the model except for changing whether to add a specific module.

*4.1. The Effect of the Attention Mechanism*

A popular attention mechanism is mainly constructed from the channel level or the spatial level, including SE, ECA and CBAM. The introduction of an attention mechanism is mostly to strengthen the target area that needs to be focused, and different attention mechanisms play different roles in the same environment. To explore the kind of attention mechanism that is most suitable for the GPAFPN, we compared two kinds of ResNet depths that were applied in the experiment. The purpose is to see that the gating function has good robustness at different depths of the network. The results are shown in Table 4. Whether in resnet34 or resnet50, the three GPAFPN structures are improved compared to the original FPN structure, and with the deepening of the network, the same method can generally show better results. When resnet34 is used as the backbone, the GPAFPN with SE as the

gating function has the most obvious improvement, and the mAP is 0.6060, which is 0.016 higher than FPN's 0.5900. This was followed by GPAFPN with CBMA and ECA as gating functions, which improved by 0.011 and 0.005, respectively. On ResNet50, which has a deeper network, the improvement of SE is still the most obvious. Its mAP reaches 0.6200, which is 0.022 higher than FPN's 0.5980. ECA and CBMA followed. After comparing with PAFPN, an interesting phenomenon is found: PAFPN has a significant improvement in accuracy than FPN regardless of the depth of the network, but the accuracy of the ECA or CBMA gating function is reduced, and only the SE gating function has a positive effect. This phenomenon also indirectly shows that not all gating functions play a positive role. Although the gating function can filter noise, it will also filter out useful information. In summary, the GPAFPN with SE structure is the most suitable for the experiments in this paper, and its accuracy improvement effect is the most obvious.

**Table 4.** The performance of different attention mechanisms on the NWPU VHR-10 dataset.

| Backbone | Method | mAP |
|---|---|---|
| Resnet-34 | FPN | 0.5900 |
| | PAFPN | 0.6030 |
| | PAFPN+ECA | 0.5950 |
| | PAFPN+CBMA | 0.6010 |
| | PAFPN+SE | **0.6060** |
| Resnet-50 | FPN | 0.5980 |
| | PAFPN | 0.6120 |
| | PAFPN+ECA | 0.6100 |
| | PAFPN+CBMA | 0.6080 |
| | PAFPN+SE | **0.6200** |

### 4.2. The Top Layer Derived from Different Feature Layers

An additional feature layer can be obtained by downsampling from the original feature layer. The classic structure is obtained by downsampling the top layer $X_1^b$, and this experiment adds an additional SSAC operation. Additionally, the performance under different original characteristic layers is explored, and the specific experimental results are shown in Table 5. It is observed that the addition of the SSAC structure does not necessarily activate the incentive effect. It has an inhibitory effect on the $X_1$ layer, while it has an incentive effect on $X_2$ and $X_3$. Among them, SSAC is used on $X_2$, and the best results are achieved on mAP, AP_50 and AP_75.

**Table 5.** The top-level performance derived from different feature maps.

| Feature Level | SSAC | mAP | $AP_{50}$ | $AP_{75}$ |
|---|---|---|---|---|
| $X_b^1$ | | 0.6200 | 0.9210 | 0.6800 |
| $X_b^1$ | ✓ | 0.6180 | 0.9190 | 0.6820 |
| $X_b^2$ | | 0.6170 | 0.9200 | 0.6820 |
| $X_b^2$ | ✓ | **0.6220** | **0.9270** | **0.6880** |
| $X_b^3$ | | 0.6130 | 0.9210 | 0.6810 |
| $X_b^3$ | ✓ | 0.6170 | 0.9210 | 0.6740 |

### 4.3. Comparison of Other Models

We use the same dataset to compare mainstream models, including Hyper [29], RICA [30], CA-CNN [9], COPD [31], RICNN [25], Fast-RCNN [32] and Faster-RCNN [26]. The mAP of our proposed GPANet is 0.927, which is state-of-the-art when compared with the above models. Analysis of the reasons found that the main point is that the accuracy of several types of small targets is slightly improved and the accuracy of other categories is seriously reduced. The specific experimental results are shown in Table 6. The observation data shows that the four categories of storage tank, vehicle, harbor and bridge are the best

in all models; in particular, the accuracy improvement of harbor and bridge is more obvious. The best models improved by 0.034 and 0.029, respectively. Many other models include the FPN structure, but the biggest difference in this article is that the GPAFPN structure includes a gating mechanism. We pass the visualization of five feature layers, as shown in Figure 6: a1 represents the original image; a2 represents the ground truth; a3 represents the prediction effect map of GPAFPN; a4 represents the prediction effect map of FPN; b and c represent the heat map of each feature layer of GPAFPN and FPN, respectively; and the red ellipse in a3 and a4 Indicates the difference between the two. Judging from the results in the red oval, GPAFPN detected one more bridge and one storage tanks than FPN. Observing the feature layers of b and c, it was found that the most targets were detected in the P2 feature layer. Among them, less bridges can be detected in the P6 layer of GPAFPN, and the effect of P6 detection is also more accurate than the target extracted by FPN. Figure 7 shows the transformation of the loss rate during the training process of the ablation experiment. The overall change trend of the loss rate in the four experiments is consistent, but near the end of the experiment, the combination of SE and SSAC achieves the smallest loss value, that is, the training effect is the best. The pure PAFPN is the worst training effect. In terms of the prediction results of GPAFPN by SE and SSAC, SE has a greater impact on the module.

**Table 6.** The performance of mainstream models.

| Method | mAP | Ship | Airplane | Storage Tank | Vehicle | Harbor | Tennis Court | Baseball Diamond | Bridge | Basketball Court | Ground Track Field |
|---|---|---|---|---|---|---|---|---|---|---|---|
| Hyper | 0.887 | 0.898 | 0.997 | 0.987 | 0.887 | 0.804 | 0.907 | 0.909 | 0.689 | 0.903 | 0.893 |
| RICA | 0.871 | 0.908 | 0.997 | 0.906 | 0.871 | 0.803 | 0.903 | 0.929 | 0.685 | 0.801 | 0.908 |
| CA-CNN | 0.910 | 0.906 | **0.999** | 0.900 | 0.890 | 0.890 | 0.902 | **0.997** | 0.793 | **0.909** | 0.909 |
| COPD | 0.807 | 0.817 | 0.891 | 0.973 | 0.833 | 0.734 | 0.733 | 0.894 | 0.629 | 0.734 | 0.830 |
| RICNN | 0.726 | 0.773 | 0.884 | 0.853 | 0.711 | 0.686 | 0.408 | 0.881 | 0.615 | 0.585 | 0.867 |
| Fast-RCNN | 0.827 | 0.906 | 0.909 | 0.893 | 0.698 | 0.882 | **1.000** | 0.473 | 0.803 | 0.859 | 0.849 |
| Faster-RCNN | 0.827 | 0.906 | 0.909 | 0.905 | 0.781 | 0.801 | 0.897 | 0.982 | 0.615 | 0.696 | **1.000** |
| GPANet | **0.927** | **0.921** | 0.995 | **0.993** | **0.892** | **0.924** | 0.923 | 0.995 | **0.832** | 0.846 | 0.923 |

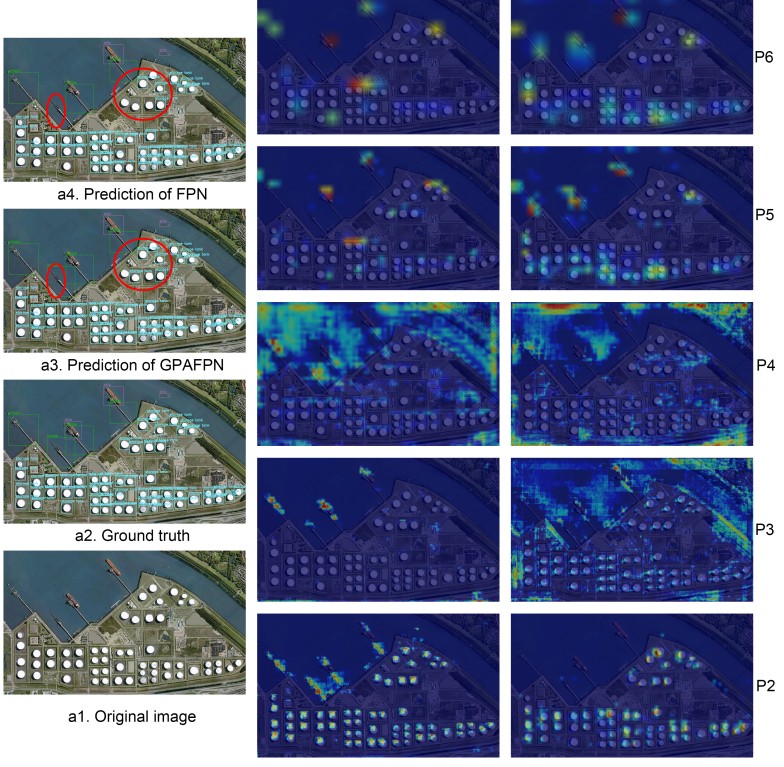

**Figure 6.** Performance of different feature layers of GPAFPN and FPN.

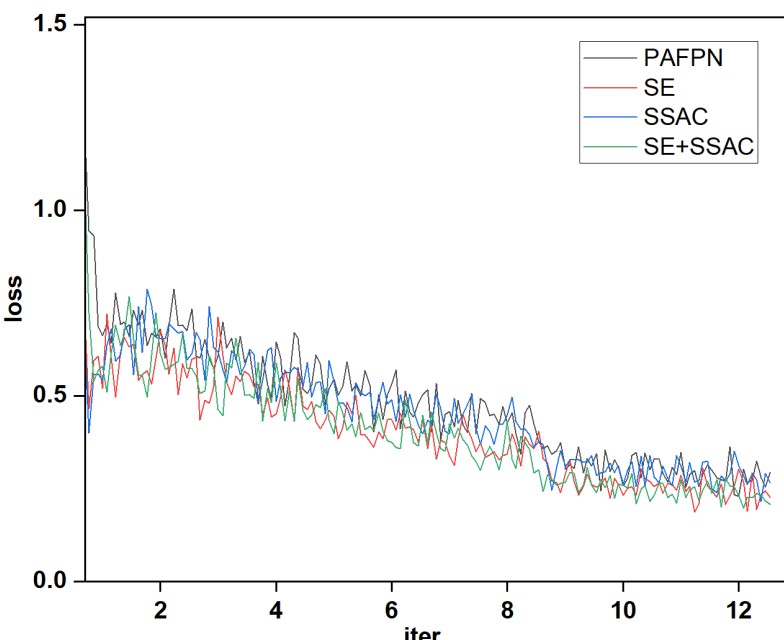

**Figure 7.** Comparison of the training loss of PAFPN, SE, SSAC and SE+SSAC during 12 training epochs.

### 5. Conclusions

In this paper, a remote sensing image target detection and classification method based on a gated path enhancement network is proposed. This method focuses on the information interaction between different layers and filters the noise information between cross-layers. Experiments show that the path enhancement method can improve the dissemination of information, and the integrated SE attention mechanism has the most obvious improvement effect on the model. SSAC selects the most suitable convolution operation to improve performance by adjusting the detection receptive field of the top layer. In future work, experiments will explore how to choose the appropriate level of fusion in information fusion instead of simply bottom-up path fusion.

**Author Contributions:** Y.Z. wrote the manuscript, designed the comparative experiments, designed the architecture and performed the comparative experiments; X.Z. and D.W. supervised the study and revised the manuscript; R.Z. revised the manuscript and gave comments and suggestions to the manuscript. All authors have read and agreed to the published version of the manuscript.

**Funding:** This work was supported by the National Natural Science Foundation of China (62103345), the Science and Technology Planning Project of Fujian Province (2020H0023, 2020J02160, 2020J01265), and the Science and Technology Planning Project of Quanzhou (2020C074), and the Science and Technology Climbing Program of Xiamen University of Technology (XPDKT18030).

**Acknowledgments:** We would like to thank the anonymous reviewers for their constructive and valuable suggestions on the earlier drafts of this manuscript. The authors also want to thank editors for their patient and meticulous work for our manuscript.

**Conflicts of Interest:** The authors declare no conflict of interest.

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
