# Peer review of "Gated Path Aggregation Feature Pyramid Network for Object Detection in Remote Sensing Images"

_remotesensing, doi:10.3390/rs14184614_

Round 1

Reviewer 1 Report (Previous Reviewer 2)

New additions and changes to the manuscript have made it much clearer.

Reviewer 2 Report (Previous Reviewer 3)

The authors have followed all my suggestions.

This manuscript is a resubmission of an earlier submission. The following is a list of the peer review reports and author responses from that submission.

Round 1

Reviewer 1 Report

The paper proposed a novel method to address the challenge of object detection in remote sensing images. In detail, the authors proposed the GPANet based on YOLOv3 to utilize semantic features from more layers. The final results show the method has very high accuracy in object detection on public earth images.

Even though the paper has a good presentation and results, there are multiple points the paper can improve. 

1. The biggest concern is the effectiveness of the method. The proposed model must be large from the paper's description, but the paper only uses far less than enough examples to train the model. The paper also does not separate validation, training, and test samples, which makes the reader hard to understand the effectiveness of the proposed model.

2. The paper failed to explain with enough theory and detail why the proposed framework can address the challenges of object detection in remote sensing images.

3. The paper's model is based on an existing popular ML model, how the author's model is warmed up with the existing model is not presented in the paper.

Reviewer 2 Report

Generally the paper is well written and easy to follow.  I do have some confusion over what the authors are trying to convey on lines 18-19 and 214. In the former I believe the main point is that remotely sensed data may have smaller targets of interest due to the larger pixel size and they may be covering larger total area.  Is this the case?  In the second case, I believe the main point is that the total number of images available may be smaller than optimal.  Perhaps this can be clarified.

Making Figure 6 larger would help convey the performance of the proposed model.  It is hard to see the bounding boxes on the targets.

Reviewer 3 Report

The manuscript presents a Gate Path Aggregate Feature Pyramid Network (GPFAPN) for object detection in remote sensing images. The main weakness of the paper is the lack of a clear justification of the novelty of the proposed technique and a proper experimental analysis that shows its advantages with respect to the state-of-the-art techniques.

The authors say in the abstract that the proposed GPAFPN structure has significant improvement compared to the FPN structure. However, the results shown in Section 4 are confusing and do not show a clear comparison between the state-of-the-art techniques (which are described in detail in Section 1) and the proposed one. The authors include a lot of information about the used dataset (which could be referred to), but they do not carry out a deep analysis of the results. Is it possible to evaluate the performance of the GPANet model by watching the pictures in Figure 6? Figure 7 is confusing and requires an explanation and an analysis. The term GPANET (used in the abstract to conclude that the proposed method is better than FPN) is used only two times in the whole paper. What is the relationship between the methods shown in Table 4 and the proposed one? What is the reason to conclude that the proposed method presents significant improvements compared to the FPN structure?